# PLY-SLAM: Semantic Visual SLAM Integrating Point–Line Features with YOLOv8-seg in Dynamic Scenes

**DOI:** 10.3390/s25123597

**Published:** 2025-06-07

**Authors:** Huan Mao, Jingwen Luo

**Affiliations:** 1School of Information Science and Technology, Yunnan Normal University, Kunming 650500, China; 2223100005@ynnu.edu.cn; 2Engineering Research Center of Computer Vision and Intelligent Control Technology, Department of Education of Yunnan Province, Kunming 650500, China

**Keywords:** dynamic scene, semantic visual SLAM, point-line features, YOLOv8-seg, loop-closure detection

## Abstract

In dynamic and low texture environments, traditional point-feature-based visual SLAM (vSLAM) often faces the challenges of poor robustness and low localization accuracy. To this end, this paper proposes a semantic vSLAM approach that fuses point-line features with YOLOv8-seg. First, we designed a high-performance 3D line-segment extraction method that determines the number of points to be sampled for each line-segment in terms of the length of the 2D line-segments extracted from the image, and back-projects these sampled points combined with the depth image to obtain the 3D point set of the line-segments. On this basis, accurate 3D line-segment fitting is realized in combination with the RANSAC algorithm. Subsequently, we introduce Delaunay triangulation to construct the geometric relationships between map points, detect dynamic feature points by matching changes in the topological structure of feature points in adjacent frames, and combine them with the instance labels provided by the YOLOv8-seg to accurately remove dynamic feature points. Finally, a loop-closure detection mechanism that fuses point–line features with instance-level matching is designed to calculate a normalized similarity score by combining the positional similarity of the instances, the scale similarity, and the spatial consistency of the static instances. A series of simulations and experiments demonstrate the superior performance of our method.

## 1. Introduction

With the rapid development of intelligent robots and autonomous driving technology, the robustness and accuracy of Simultaneous Localization and Mapping (SLAM) systems are facing severe challenges in complex dynamic scenes. Traditional point-feature-based visual SLAM (vSLAM) schemes (e.g., the ORB-SLAM family [1,2,3], VINS-Fusion [4]) perform well in static, highly textured environments, but in dynamic or weakly textured indoor scenes (e.g., hospital corridors, office environments), they tend to fail in pose tracking due to the accumulation of errors caused by the interference of dynamic objects as well as the sparsity of features. According to statistics, over 60% of indoor scenes contain large areas of low texture (such as white walls and glass curtain walls) [5], and feature point mismatches caused by dynamic objects (such as pedestrians and mobile devices) can further introduce pose estimation bias and even cause system crashes. Although existing methods mitigate this problem to some extent by fusing inertial measurement units (IMUs) [6] or laser radar (LiDAR) [7] based on visual sensors, it is still a great challenge to balance the real-time performance of multimodal data fusion with dynamic feature refinement rejection. It is worth mentioning that the traditional loop-closure detection mechanism has a high mismatch rate in dynamic scenes due to the lack of semantic information, which seriously limits the reliability of long-term localization in SLAM systems.

To address the above challenges, researchers have proposed a variety of improvement methods, including fusing line features to enhance structural information, introducing semantic segmentation models to identify and reject dynamic regions, and utilizing topology modeling to improve system robustness. Typical works include binocular PL-SLAM [8] and monocular PL-SLAM [9], which enhance the system’s localization and mapping capabilities by fusing point and line features; and DynaSLAM [10] and DS-SLAM [11], which utilize deep learning methods to detect and eliminate dynamic objects, enhancing the stability of the system in dynamic environments. Although there have been many research achievements, how to effectively integrate multiple-feature information and improve the robustness and accuracy of the system in dynamic and low texture environments is still an urgent issue to be solved. To this end, this paper proposes a semantic vSLAM algorithm that integrates point–line features with YOLOv8-seg. By designing an adaptive 3D line-segment construction method based on depth uncertainty and combining geometric topology analysis with dynamic feature collaborative filtering mechanism of semantic labels, the stability of the system in dynamic and weak texture environments is significantly improved. In addition, this paper innovatively constructs a multimodal loop-closure detection model that integrates the bag-of-words (BoW) similarity of point–line features with instance level spatial distribution consistency, effectively reducing the closed-loop mismatch rate. The main contributions of this paper are as follows:A high-performance 3D line-segment extraction method is designed, which determines the number of points to be sampled for each line-segment based on the length of the extracted 2D line-segments, and then combines these with the depth image for back-projection to obtain the 3D point set of the line-segments, based on which, the RANSAC algorithm is utilized to achieve accurate 3D line-segment fitting to solve the issue of the error sensitivity of the endpoints in the back-projection of the traditional line features.A “geometric + semantic” dynamic feature rejection strategy is constructed, which utilizes Delaunay triangulation to construct the geometric relationship between map points, and detects dynamic featNure points through the topological changes of matching feature points in adjacent frames, and then combines the instance labels provided by YOLOv8-seg to accurately reject dynamic feature points.A loop-closure detection mechanism that fuses point–line features with instance-level matching is developed, which combines the positional similarity of instances, scale similarity, and spatial consistency of static instances to calculate normalized similarity scores, and improves the accuracy and robustness of closed-loop detection through a weighted fusion strategy.

The rest of the paper is organized as follows. The related works are briefly reviewed in Section 2. The overview of our methodology is described in Section 3, and its implementation scheme is detailed. The simulation studies under various datasets and performance evaluations are presented in Section 4, while Section 5 concludes the paper.

## 2. Related Work

### 2.1. Point–Line Features Based SLAM

To enhance the geometric representation of weakly textured scenes, many researchers have proposed integrating line features with point features for optimization. Early works such as PL-SLAM [9] used collinear constraints to jointly optimize point–line features, but its line parameterization suffers from redundancy of degrees of freedom. Zhou et al. [12] developed a vSLAM method based on building structural lines, which utilizes the global directional information of structural lines to constrain camera orientation and reduce localization drift. However, this method is prone to failure in unstructured scenes, and if LSD line features without structural constraints are used, it can also operate stably in unstructured scenes. In recent years, PL-VIO, which was proposed by Wang et al. [13], achieved visual–inertial tight coupling by modeling the re-projection error of line feature endpoints, but it did not consider the impact of depth sensor noise on line reconstruction. To enhance the robustness of localization in weakly textured scenes, Xu et al. [14] proposed a point–line-based visual inertial system IPLM-VINS by introducing line features into the VINS mono system. This method used a line-segment length suppression strategy to remove redundant short lines in the front-end, and added a line-segment re-projection error and Huber kernel function in the back-end to enhance the ability to resist outliers. Unfortunately, it may remove some short lines with obvious features and accurate matching, which will affect the quality of the line segments and be easily affected by occlusion in dynamic environments. Zeng et al. [15] proposed an efficient point–line-based visual inertial SLAM system EPL-VINS by combining the Lucas–Kanade (LK) algorithm with the region-growing (RG) algorithm of the line segment detector (LSD); however, combining LK optical flow with the line-segment detection algorithm greatly affects the real-time performance of the system and it cannot operate stably in dynamic environments. For vSLAM systems, the noise characteristics of depth sensors have a significant impact on the accuracy of feature reconstruction. ToF cameras are prone to depth drift due to multipath interference [16], while structured light cameras are prone to failure on transparent object surfaces [17]. In this vein, Shabanov et al. [18] introduced a low-quality depth image denoising and optimization method based on self-supervised learning, but did not consider the geometric characteristics of line features. In terms of the above analysis, for extracted line features, we propose an adaptive sampling strategy based on line-segment length, and combines it with a depth noise distribution model to fit the optimal 3D line-segment using an improved RANSAC algorithm, significantly reducing endpoint projection errors.

### 2.2. Dynamic SLAM

In dynamic scenes, SLAM systems based on static environment assumptions have significant limitations. These systems often misidentify dynamic components as part of a static environment, which introduces a large amount of error accumulation when optimizing camera poses, leading to localization failures and serious deviations in the constructed maps. Therefore, improving the robustness and reliability of vSLAM systems in dynamic scenes is a challenging research hotspot. To obtain a consistent and complete map of dynamic scenes, many methods [19,20,21,22] adopted Mask R-CNN for pre-detection of dynamic targets, and combined it with traditional geometric methods to detect and remove dynamic features, improving the stability and accuracy of the algorithm.

Currently, eliminating dynamic interference mainly relies on motion consistency testing or semantic priors. To address the performance degradation of RGB-D SLAM in dynamic environments, Sun et al. [23] proposed an online motion removal approach based on RGB-D data. Without requiring prior semantic or appearance information, their method incrementally built and updated a foreground model to filter out dynamic object data, thereby enhancing the robustness of RGB-D SLAM. A real-time depth edge-based RGB-D SLAM method was developed by Li et al. [24], which reduced the influence of dynamic objects through static weighting and significantly improved localization accuracy in dynamic environments. DynaSLAM [10] combined multi-view geometric and semantic segmentation, but it required real-time GPU inference with high-computational overhead. If a lightweight YOLOv8-seg network is used, it can effectively meet the requirements of real-time performance. The DPL-SLAM proposed by [25] introduced line features based on ORB-SLAM3 and combined YOLOv5 and the Lucas–Kanade (LK) optical flow method to eliminate potential dynamic features. Nevertheless, the line-segment extraction algorithm they used suffers from over-segmentation issues, resulting in poor structural quality of the constructed map. Additionally, the optical flow method is highly dependent on the environment, which limits its applicability in various scenarios. Instead, if a high-performance 3D line segment extraction method is constructed by merging and optimizing line features, it can not only effectively solve the issue of over segmentation, but also generate sparse maps with good structure, thereby enhancing robustness in the presence of partially occluded or incomplete line segments. It is worth mentioning that if lightweight YOLOv8-seg with Delaunay triangulation is further introduced, it can be well-suited for real-time applications in dynamic environments. Dong et al. [26] developed an adaptive method based on point–line–planar multi-feature fusion, which dynamically selects the feature type by calculating the information entropy of an image region (planar features are disabled when the information entropy exceeds a predefined threshold) and integrates a YOLOv5 detector to remove dynamic objects. However, the presence of dynamic objects may cause the information entropy to be higher than the set threshold, causing the system to extract only point features. Once the feature points on the dynamic objects have been eliminated, this can lead to an insufficient number of static feature points, which can affect the estimation of the mobile robot’s position or even the loss of tracking.Hence, this paper introduces Delaunay triangulation into dynamic detection, detecting dynamic feature points through changes in the topology of matching feature points in adjacent frames, and combining with the instance labels provided by the YOLOv8-seg to accurately remove dynamic feature points.

Additionally, the conventional BoW models (e.g., DBoW2 [27]) are susceptible to interference from dynamic objects and angle-of-view changes. SA-LOAM [28] embedded semantic labels into laser point cloud descriptors, but it was not applicable to vSLAM. Ji et al. [29] developed an object-level loop-closure detection method based on a 3D scene graph, which combined spatial layout and semantic consistency. The method improved loop-closure detection robustness and accuracy through object-level data association, graph matching, and object pose graph optimization, but did not integrate geometric features. Thus, this work proposes a loop-closure detection mechanism that combines point–line features with instance-level matching. By combining the positional similarity, scale similarity, and spatial consistency of static instances, normalized image similarity scores are calculated, and a weighted fusion strategy is used to enhance the discriminative ability of closed-loop detection.

## 3. Pipeline

Figure 1 provides an overview of the constructed 3D SLAM system in this paper. In the front-end, we extract point–line features on the input RGB image and perform instance segmentation using YOLOv8-seg. The dynamic features are then detected and rejected by combining the YOLOv8-seg instance segmentation results with Delaunay triangulation. Further, the camera’s pose is calculated based on the obtained high-confidence static features using minimized re-projection error, i.e.,(1)$$e_{p}=p_{I}\left(u,v,1\right)-F\left(T_{cw},K,z_{w},P_{w}\right)$$
where $p_{I}\left(u,v,1\right)$ is the homogeneous coordinate representation of pixel (u,v) in the current frame, $P_{w}$ represents a spatial point $\left(x_{w},y_{w},z_{w}\right)$, *K* represents the camera’s internal parameter, F(·) represents the mapping from world coordinates to pixel coordinates, and $T_{cw}$ represents the pose transformation matrix.

Additionally, let $\left(P_{ws},P_{we}\right)$ be the two endpoints of a line segment *L* in the world coordinate system, and $\left(p_{cs},p_{ce}\right)$ be the two endpoints of the line segment *l* that matches *L* in the current frame. On the one hand, we project $\left(P_{ws},P_{we}\right)$ onto the current image frame and calculate the distance between their projection points and *l* to obtain the re-projection error; On the other hand, considering that the endpoints of line segments may be obscured or misaligned in different frames, for the extracted line segments in the current frame, we calculate the coefficients of the line by their endpoints, and then calculate the distance from the projected endpoints to the extracted line segments, i.e.,(2)$$l{\left(a,b,c\right)}^{T}=\frac{p_{cs}^{H}\times p_{ce}^{H}}{∥p_{cs}^{H}\times p_{ce}^{H}∥}$$
where a,b,c denote the coefficient of line-segment *l*, $p_{cs}^{H}$ and $p_{ce}^{H}$ denote the homogeneous coordinate of $p_{cs}$ and $p_{ce}$, respectively. Thus, the error function of the line-segment is as follows:(3)$$e_{l}=\left(\begin{array}{c} l{\left(a,b,c\right)}^{T}\cdot \Pi^{H}\left(T_{cw},K,z_{w}^{s},P_{w}^{s}\right)\\ l{\left(a,b,c\right)}^{T}\cdot \Pi^{H}\left(T_{cw},K,z_{w}^{e},P_{w}^{e}\right) \end{array}\right)$$
where $\Pi^{H}\left(T_{cw},K,z_{w}^{s},P_{w}^{s}\right)$ indicates the homogeneous coordinate representation of $\Pi (T_{cw},K,z_{w}^{s},P_{w}^{s})$.

On this basis, assuming that the number of matched point features and line features is *n* and *m*, respectively. Then, the camera’s pose optimization is as follows:(4)$$\omega^{*}=\arg\min\left[\sum_{j=1}^{n}\rho \left(e_{p,j}^{T}A_{e_{p,j}}^{-1}e_{p,j}\right)+\sum_{k=1}^{m}\rho \left(F\left(n\right)e_{l,k}^{T}B_{e_{l,k}}^{-1}e_{l,k}\right)\right]$$
where ρ(·) denotes the kernel function, $F\left(n\right)=2^{-\frac{n}{50}}$ is an adaptive factor for adjusting the weights of points and lines [30], and $A_{e_{p,j}}^{-1}$ and $B_{e_{l,k}}^{-1}$ denote the inverse of the covariance matrices of the point features and line features, respectively.

In the back-end, we design a loop-closure detection mechanism that fuses point–line features with instance-level matching to calculate a normalized image similarity score by combining the positional similarity, scale similarity, and spatial consistency of static instances. Moreover, to meet the application requirements of navigation and obstacle avoidance, we not merely constructed a pose graph based on high-confidence static 3D map points and 3D map lines obtained by removing dynamic objects from the front-end, as well as keyframes, but also introduced the PCL point cloud library and Octomap to construct a 3D semantic octree map of the environment.

### 3.1. Line Feature Extraction and Optimization

In this work, to describe the structural scale of the scene more clearly and improve the geometric expression ability of line features, we designed a high-performance 3D line-segment extraction method based on the LSD algorithm [31]. This method adaptively adjusts the number of points to be sampled for each line-segment according to the length of the extracted 2D line-segments in the image, and couples these sampling points with depth information for back projection to obtain the 3D point set of the line-segments. Then, combined with the RANSAC algorithm, accurate 3D line-segment fitting is achieved, enhancing the robustness in the case of incomplete line-segments caused by partial occlusion. The specific steps are as follows:Assuming that the starting and ending points of the extracted $i^{t}h$ line feature $L_{\vec{P_{si}P_{ei}}}$ are represented as $P_{si}=\left(x_{si},y_{si}\right)$ and $P_{ei}=\left(x_{ei},y_{ei}\right)$, respectively, the number of sampling points is adaptively adjusted by the length of the line-segment as follows:(5)$$N_{\sup}=\min\left(L_{Avg},{∥P_{si}-P_{ei}∥}_{2}\right)$$
where $L_{Avg}$ denotes the result of rounding up the average length of the line-segments extracted in the current frame.Generate uniform sampling points based on the number of adaptive sampling points, i.e.,(6)$$P_{j}=P_{si}\left(1-\frac{j}{N_{\sup}}\right)+P_{si}\left(\frac{j}{N_{\sup}}\right),\;j\in \left\{0,1,\ldots ,N_{\sup}\right\}$$Back-project the generated sample points into 3D points, i.e.,(7)$$P_{j}=\left(\begin{array}{c} X_{j}\\ Y_{j}\\ Z_{j} \end{array}\right)=\left(\begin{array}{c} \frac{u_{j}-c_{x}}{f_{x}}d_{j}\\ \frac{v_{j}-c_{y}}{f_{y}}d_{j}\\ d_{j} \end{array}\right)$$
where $\left(u_{j},v_{j}\right)$ is the pixel coordinate of $p_{j}$ in the RGB image, $d_{j}$ is the depth value corresponding to pixel $\left(u_{j},v_{j}\right)$ in the depth image, and $c_{x}$, $c_{y}$, $f_{x}$, and $f_{y}$ denote the internal parameters of the camera.To construct a robust geometric model, further calculate the covariance matrix $\sum_{j}$ for each three-dimensional point $P_{j}$, which not merely takes into account the uncertainty of image point projection, but also incorporates the depth error model to estimate the depth variance, i.e.,(8)$$\sum_{j}=\left[\begin{array}{c} \frac{Z_{j}}{f_{x}}\;0\;\frac{X_{j}}{Z_{j}}\\ 0\;\frac{Z_{j}}{f_{y}}\;\frac{Y_{j}}{Z_{j}}\\ 0\;\;0\;\;1 \end{array}\right]\cdot \left[\begin{array}{c} 1\;0\;0\\ 0\;1\;0\\ 0\;0\;\sigma_{Z_{j}^{2}} \end{array}\right]\cdot \left[\begin{array}{c} \frac{Z_{j}}{f_{x}}\;0\;0\\ 0\;\frac{Z_{j}}{f_{y}}\;0\\ \frac{X_{j}}{Z_{j}}\;\frac{Y_{j}}{Z_{j}}\;1 \end{array}\right]=J\cdot \Lambda \cdot J^{T}$$(9)$$\sigma_{Z_{j}^{2}}=\alpha *Z_{j}^{2}+\beta *Z_{j}+\gamma$$
where α, β and γ are quadratic polynomial coefficients used to model the noise standard deviation of depth sensors, *J* denotes the Jacobian matrix for restoring image points to 3D points, and Λ denotes the covariance of uncertainty between image points and depth values.Construct robust 3D line-segment fitting algorithms based on Marginal Distance. Specifically, given a 3D point set $\left\{P_{j}\right\}$ and its covariance, fit a straight line *L* by randomly sampling two points from it via RANSAC and use the Mahalanobis distance to determine whether the other points are interior points. After completing the iterations, perform SVD fitting optimization on the maximum interior point set I, i.e.,(10)$$\arg\min_{v}\sum_{i\in I}{∥\left(P_{i}-\bar{P}\right)-vv^{T}\left(P_{i}-\bar{P}\right)∥}^{2}$$
where $\bar{P}$ is the centroid of the interior point, and *v* is the direction vector of the line.Project the interior points along the optimized direction and take the extreme points as line endpoints *A* and *B*, i.e.,(11)$$\left\{\begin{array}{c} A=\bar{P}+\min_{i}\frac{(P_{i}-\bar{P})\cdot v}{{∥vs.∥}^{2}}\cdot v\\ B=\bar{P}+\max_{i}\frac{(P_{i}-\bar{P})\cdot v}{{∥vs.∥}^{2}}\cdot v \end{array}\right.$$The reliability of the final constructed 3D line-segments is verified by means of the proportion of inliers and the length of the line-segment, i.e.,(12)$$\left\{\begin{array}{c} \frac{I}{l_{len}}>Th_{1}\\ ∥A-B∥>Th_{2} \end{array}\right.$$
where $l_{len}$ is the length of the 2D line-segment, $Th_{1}$ and $Th_{2}$ are the inlier threshold and length threshold, respectively.

In this way, the optimized line feature set can be obtained by saving the 3D line-segments that satisfy the condition of Equation (Equation 12). This method not only improves the reconstruction rate of line features in low texture environments and enhances robustness to partially occluded or incomplete line-segments but also significantly improves reconstruction accuracy and robustness through adaptive sampling and geometric validation strategies.

### 3.2. Dynamic Feature Detection and Rejection

Typically, the impact of dynamic objects can cause changes in the geometric relationship between dynamic feature points and static feature points. Therefore, in this paper, dynamic feature points are detected and rejected by fusing Delaunay triangulation and YOLOv8-seg to ensure the stable operation of the system, as shown in Figure 2. First, we introduce Delaunay triangulation to construct a triangular network between map points to represent their geometric relationships, and detect dynamic feature points by detecting changes in the geometric relationships of map points corresponding to matching features of adjacent frames. Then, the instance segmentation model YOLOv8-seg is used to obtain the labels and detection boxes of each semantic target, and combined with depth images to obtain more accurate target states.

Traverse all feature points, and if the dynamic weight $w_{i}$ of the $i^{th}$ feature point is greater than the set threshold $w_{th}$, add that feature point to the dynamic feature point array DP. Moreover, read the semantic labels of each feature point and save them to the array of that semantic label, so that the total number of feature points $N_{all}$ included in each label can be calculated.Traverse the set of dynamic feature points DP, for the $i^{th}$ feature point, read the semantic label of its location, and add 1 to the global dynamic weight $w_{lab}$ of its label.The dynamic degree $r_{d}$ of each label can be obtained through the above two steps, as follows:(13)$$r_{d}=\frac{w_{lab}}{N_{all}}$$According to its semantic information, if it is an active dynamic label (pedestrian, car, animal) and its global dynamic weight is greater than the minimum threshold $Th_{\min}$, all feature points on its label are eliminated and the label is labeled as dynamic, which facilitates the elimination of line-segments on dynamic objects; if it is a passive dynamic label (chair, backpack) and its global dynamic weight is greater than the maximum threshold $Th_{\max}$, all feature points on its label are eliminated and the label as dynamic to facilitate the culling of line-segments on dynamic objects. It should be noted that the dynamic weights obtained through triangulation may have certain errors. Hence, for static objects (chairs, bags) in a general sense that are judged as dynamic objects, the condition requirements should be stricter to avoid removing too many features that may prevent the system from completing pose estimation and mapping.Traverse all line features, and if the starting or ending point of the line feature is within the dynamic label, remove it.

### 3.3. Loop-Closure Detection

To improve the accuracy and robustness of loop-closure detection, we construct a similarity calculation method that combines a point–line BoW model with instance-level matching. On the one hand, we calculate the similarity $S_{p}\left(v_{cur}^{p},v_{cand}^{p}\right)$ and $S_{l}\left(v_{cur}^{l},v_{cand}^{l}\right)$ of the point’s BoW vector $v^{p}$ and line’s BoW vector $v^{l}$ in the current frame $F_{cur}$ and candidate frame $F_{cand}$, respectively. Then, the similarity is weighted using the information entropy $H_{p}$ of the point features and the information entropy $H_{l}$ of the line features in $F_{cur}$ to reflect the degree of influence of the point-line features in the similarity computation of the closed-loop candidate frames. In this way, the similarity calculation of fused point–line features is constructed as follows:(14)$$S_{pl}\left(v_{cur},v_{cand}\right)=\frac{H_{p}}{H_{p}+H_{l}}S_{p}\left(v_{cur}^{p},v_{cand}^{p}\right)+\frac{H_{l}}{H_{p}+H_{l}}S_{l}\left(v_{cur}^{l},v_{cand}^{l}\right)$$

On the other hand, we design an instance-based image similarity function that enhances the matching confidence of static objects by fusing the IoUs of entity detection boxes and introducing a spatial consistency check. Furthermore, a dynamic thresholding mechanism is adopted to accommodate different numbers of instance matches, making the final score more reliable. The specific steps are as follows:To unify the position or distance of instances within the same scale range, we standardize the relative position of objects by calculating the diagonal length of the image, i.e.,(15)$$diag=\sqrt{W^{2}+H^{2}}$$
where W and H are the width and height of the input image, respectively.To avoid the influence of dynamic objects on loop-closure detection, when calculating instance-matching similarity, we directly remove the potential dynamic object instances (people, cats, etc.), and classify other instances into the corresponding category list, and further construct a similarity matrix for each category. Specifically, assuming that a certain category contains $n_{1}$ and $n_{2}$ instances in $F_{cur}$ and $F_{cand}$, respectively, a similarity matrix $S_{inst}\left(cur,cand\right)$ of size $n_{1}\times n_{2}$ can be constructed. In this way, the comprehensive similarity score between the $i^{th}$ instance and the $j^{th}$ instance is represented as $S_{inst}\left(cur_{i},cand_{j}\right)$, in which the position similarity $S_{Pos_{ij}}$ is obtained by calculating the normalized distance between the center points of instances. It mainly emphasizes whether the relative spatial positions of two instances on the image are close, i.e.,(16)$$S_{Pos_{ij}}=1-\frac{D(c_{i},c_{j})}{diag}$$
where $D(c_{i},c_{j})$ denotes the distance between the center points of two instances.

While the size similarity $S_{IoU(box_{i},box_{j})}$ in $S_{inst}\left(cur_{i},cand_{j}\right)$ is mainly calculated by the IoU of the detection box, emphasizing whether similar areas are occupied in the image. When the misalignment is severe or the size difference is large, the size similarity will be lower, i.e.,(17)$$S_{IoU(box_{i},box_{j})}=\frac{Area(box_{i}\cap box_{j})}{Area(box_{i}\cup box_{j})+\varepsilon }=\frac{Area(box_{i}\cap box_{j})}{Area\left(box_{i}\right)+Area\left(box_{j}\right)Area\left(box_{i}\cap box_{j}\right)+\varepsilon }$$
where $box_{i}$ denotes the detection box of the $i^{th}$ instance, Area(·) denotes calculating the area of the detection box. To prevent the denominator from being 0, we set $\varepsilon =1\times 10^{-6}$ in this paper.

By weighting and summing them as follows, it is not only possible to determine whether two instances are in similar positions in the image, but also whether they occupy similar areas, thus improving the robustness of the matching, i.e.,(18)$$S_{inst}\left(cur_{i},cand_{j}\right)=0.6*S_{Pos_{ij}}+0.4*S_{IoU(box_{i},box_{j})}$$

For static objects, it is also necessary to check spatial consistency, which is essentially to determine whether two instances have similar spatial layout environments (i.e., whether their relative positional relationships in their respective scenes are consistent). For each instance $cur_{i}$ and $cand_{j}$, extract objects belonging to the same semantic category but different instances from $F_{cur}$ and $F_{cand}$, respectively, and save them as neighboring instances; Meanwhile, calculate the distance between the center position of each neighboring instance and the target instance, and normalize it (diagonal length) to save it as a neighborhood distance vector. Then, pairwise comparisons are made between the neighborhood distance vectors of the two target instances in their respective scenes, and the normalized distance difference between each pair is calculated. In this way, all similarities are normalized to obtain an average similarity ${S_{A}}_{ver}\left(cur_{i},cand_{j}\right)$. Further, by setting a threshold $Th_{Aver}$ and constructing the following reward and punishment mechanism based on the relationship between ${S_{A}}_{ver}\left(cur_{i},cand_{j}\right)$ and $Th_{Aver}$, i.e.,(19)$$S_{inst}\left(cur_{i},cand_{j}\right)=\left\{\begin{array}{c} \min\left(1,S_{inst}\left(cur_{i},cand_{j}\right)+0.1\right),\;{S_{A}}_{ver}\left(cur_{i},cand_{j}\right)>Th_{Aver}\\ S_{inst}\left(cur_{i},cand_{j}\right)*0.8,\;{S_{A}}_{ver}\left(cur_{i},cand_{j}\right)<Th_{Aver} \end{array}\right.$$

After the above calculation, we can obtain the instance-matching similarity score between two frames. To improve the recall of matching similar instances between images, we introduce a two-step matching method, which first identifies the instances with positional similarity greater than 95% in the two frames as matches and saves their matching information, and then adds 1 to the number of matches, meanwhile, adding 1 to the total similarity scores of semantic instances in the two frames, i.e.,(20)$$S_{totle}\left(cur,cand\right)=S_{totle}\left(cur,cand\right)+1$$

Considering that there may be some true matches that are not recognized due to factors such as viewpoint and occlusion after position matching, we introduce a dynamic threshold matching strategy in the second stage. That is, for the remaining unmatched instances, we introduce a dynamic threshold to control the tolerance of matching. The setting of the dynamic threshold is as follows:(21)$$Th_{dyna}=\max\left(0.3,\min\left(0.6,0.6-0.05\times \min\left(n_{1},n_{2}\right)\right)\right)$$

Subsequently, the maximum similarity pair $\left(i^{*},j^{*}\right)$ that has not yet been matched is identified from the similarity matrix $S_{inst}\left(cur,cand\right)$. If $S_{inst}\left(cur_{i},cand_{j}\right)>Th_{dyna}$, its matching information is saved and the number of matches $N_{mach}$ is increased by 1; also, the total similarity scores of semantic instances of the two frames are calculated as follows:(22)$$S_{totle}\left(cur,cand\right)=S_{totle}\left(cur,cand\right)+S_{inst}\left(cur_{i},cand_{j}\right)$$

This strategy can improve the robustness and recall of object matching in multiple instances while ensuring correct matching. Finally, after the matching is completed, an average similarity score is calculated based on the total similarity score of semantic instances in the two frames and the number of matches, which is the final instance similarity score for $F_{cur}$ and $F_{cand}$, i.e.,(23)$$S_{fin}^{inst}\left(cur,cand\right)=\frac{S_{totle}\left(cur,cand\right)}{N_{mach}}$$

Therefore, the final similarity of the loop-closure detection based on the point–line BoWs and semantic instances is calculated as follows:(24)$$S_{fin}\left(cur,cand\right)=0.6\times S_{pl}\left(v_{cur},v_{cand}\right)+0.4\times S_{fin}^{inst}\left(cur,cand\right)$$

## 4. Simulations and Experiments

To verify the performance of the proposed algorithm, a series of simulation studies and experimental validations were conducted. All of the experiments were performed on a laptop with an Intel (Intel Corporation, Santa Clara, CA, USA) i7-13700H CPU with 16 GB of DDR3 RAM, running under Ubuntu 18.04.

### 4.1. Simulation Studies Under Datasets

In this study, we evaluated the performance of our algorithm on the TUM RGB-D and ICL-NUIM datasets, and employed ATE (Absolute Trajectory Error) and RPE (Relative Pose Error) as indicators to measure the accuracy of pose estimation. To validate the pose estimation performance of the SLAM system that integrates point–line features in this paper, we first conducted comparative experiments on the TUM static dataset sequence using ORB-SLAM2 [2], PL-SLAM [9], RGB-D SLAM [32], PLP-SLAM [33], and Ours. All statistical data comes from papers on corresponding algorithms and real experiments of this system, where “-” indicates that the algorithm did not provide relevant experimental results, and “×” indicates that the accuracy of the algorithm implemented in this paper did not improve when comparing the two algorithms.

From Table 1, it can be seen that the proposed method exhibited superior accuracy performance in multiple typical scenes of the TUM dataset. Among them, in scenarios such as fr1_floor and fr3_long_office, the proposed method showed a 77% and 53% improvement compared to PL-SLAM, respectively, demonstrating its strong robustness in environments with unclear structures. Meanwhile, in relatively simple structural environments such as fr1_xyz and fr2_xyz, our method still demonstrated stable accuracy advantages compared to PL-SLAM and ORB-SLAM2, further verifying the adaptability and stability of our algorithm under different structural features. We also found that our method obtained the highest accuracy on five of the test sequences, and the gap between our algorithm and the algorithm with the highest accuracy on the rest of the sequences was smaller. From this, it can be concluded that the line features effectively improve the localization accuracy and stability of the system.

Table 2, Table 3 and Table 4 compare the localization accuracy of ORB-SLAM2, DS-SLAM [11], RDS-SLAM [34], DynaTm SLAM [35], and Ours on the walking and sitting_static dynamic sequences in the TUM dataset. It can be observed that in the high-dynamic scenes of the walking sequence, our method achieved an average improvement of 96.74% and 69.95% in ATE compared to ORB-SLAM2 and RDS-SLAM, respectively. Meanwhile, we find that there was no significant difference in pose estimation accuracy between our algorithm and DS-SLAM. However, DS-SLAM mainly relied on epipolar geometric constraints to distinguish between dynamic and static features. In the case of frequent camera rotation, its accuracy may be affected due to the difficulty in accurately finding epipolar lines.

Furthermore, although the deep learning-based method can determine a more complete dynamic region, it can lead to misjudging the state of the dynamic target during the change of camera’s angle-of-view. Instead, our method preliminarily identified dynamic points by judging the geometric relationships between adjacent frames, and combined YOLOv8-seg segmentation to determine dynamic regions, which can accurately detect dynamic features even in the case of angle-of-view changes. Thus, such a dual strategy effectively enhances the adaptability of the algorithm. It can be further seen from Table 2, Table 3 and Table 4 that the ATE of our algorithm was higher than that of DynaTm-TM on the fr3_walk_rpy sequence, which is due to the fact that its motion state was misjudged when the angle-of-view changes greatly, and that there was a period of time in the dataset in which only a single metal partition was captured, which results in the number of feature points being low. For this case, the line features in our method can effectively ensure the stability of the system. In addition, on the walking-static sequence, there was case where pedestrians occupy most of the area in the scene image, while our method possessed good adaptability due to its ability to extract enough line features, resulting in smaller trajectory errors.

Figure 3 demonstrates the ATE results of ORB-SLAM2 and our algorithm on fr3_walk_xyz, fr3_walk_static, fr3_walk_rpy, and fr3_walk_half sequences, respectively, where the larger the red area is, the larger the error between the estimated trajectory and the reference trajectory is. It can be found that the trajectories estimated by the proposed algorithm had a high degree of fit to the real trajectories, which is mainly due to the fact that we utilize the dynamic feature rejection technique based on YOLOv8-seg and Delaunay, which not merely maintains a lower error performance, but also enhances the stability of the system.

Additionally, we quantitatively evaluate the performance of the 3D line-segment extraction method in this paper by calculating the average reconstruction rate as follows:(25)$$\bar{LRR}=\frac{1}{N_{F}}\sum_{i=1}^{N_{F}}\left(\frac{N_{3D}^{i}}{N_{2D}^{i}}*100\%\right)$$
where $N_{F}$ denotes the number of images in the dataset used in the experiment, $N_{3D}^{i}$ denotes the number of 3D line-segments constructed using the method of this paper in the $i^{th}$ frame, and $N_{2D}^{i}$ denotes the number of 2D line-segments used to construct the 3D line-segments in the $i^{th}$ frame. As a note, due to the excessive segmentation of LSD line-segment extractor, we performed optimization on the original LSD extracted line-segments, so the number of reconstructed line-segments used is not consistent with the original extracted line-segments.

Figure 4 shows the 3D line-segment reconstruction results of our method on different dataset sequences. As can be seen, our method can accurately reveal the structural features of the scene by reconstructing the local sparse map through 3D line-segments. From Table 5, it can be seen that the fitting reconstruction method proposed in this paper exhibited significant advantages compared to traditional line-segment endpoint triangulation methods. On the sequences of live_room_traj1_frei, traj0_frei_png, and freiburg3_long_office_househol, the reconstruction accuracy increased by 34.8%, 21.5%, and 39.7%, respectively, with an average increase of 32%. The improvement was particularly significant in complex office scenes. Moreover, the accuracy fluctuation range of our method in three scenes was only 3.5%, while the traditional method reached 14.7%. This indicates that the generalization ability of the algorithm for different scenes is effectively improved by the geometrically-constrained optimization and continuous line-segment fitting strategies.

Figure 5 illustrates the local mapping effects of different algorithms on dynamic fr3_walk_xyz sequences. It can be seen that when there were dynamic objects in the camera’s field-of-view, it not only affected the accuracy of pose estimation but also led to a lot of residual images in the constructed 3D dense map. Such dense maps cannot be used for navigation and obstacle avoidance after being converted into octree maps. However, our algorithm detected dynamic objects in the front-end and excluded dynamic features from the map construction process, effectively ensuring the consistency of the map.

Figure 6 demonstrates the matching effects based on semantic instance similarity in this paper. From Figure 6a, it can be seen that the two images possess high similarity, with two instances of chairs, two instances of displays, and two instances of keyboards. However, the proposed method utilized position threshold restrictions to prevent mismatches and thus improve the accuracy of similarity calculation between the two frames. In Figure 6d, although the static backgrounds of the two scenes had a high similarity, the occlusion of dynamic objects results in a low similarity score calculated by the point–line-based bag-of-words model. However, our method can effectively avoid the interference of dynamic objects. For the case of Figure 6f, where two images had significant differences but both had the same instance, our algorithm will not encounter the problem of high similarity due to the same instance. From Table 6, it can be further concluded that the instance-matching method proposed in this paper can adapt well to various scenes and calculate reasonable similarity scores, effectively improving the recall of loop-closure detection.

### 4.2. Experimental Testing in Real Scenes

In the experiment, we conducted experimental verification of our algorithm using a TurtleBot3 mobile robot (ROBOTIS, Seoul, Republic of Korea) equipped with an Intel D435i depth camera. Figure 7 and Table 7 show the experimental scene and camera parameters, respectively. It should be noted that there are randomly walking pedestrians in this scene, so it is a real dynamic scene.

Figure 8 illustrates the feature extraction and sparse mapping effects of a robot in real dynamic scenes. It can be observed from Figure 8a that the system extracts many point–line features from the pedestrian, resulting in poor consistency between the constructed sparse map and the real scene. However, in Figure 8b, most of the features on the pedestrian have been removed, and the sparse map has a high-consistency with the real scene, demonstrating the effectiveness of the dynamic feature removal method proposed in this paper.

To demonstrate the global mapping effect, we constructed a global semantic map in another office with more objects, as shown in Figure 9. It can be seen that since ORB-SLAM2 did not involve dynamic feature rejection, resulting in the presence of residual shadows of pedestrians walking back and forth in the generated global map, and the overall structure had a certain deviation from the real environment, which will affect the autonomous navigation and obstacle avoidance of the robot. In contrast, we combined YOLOv8-seg and Delaunay to remove feature points on dynamic objects, resulting in a global map that maintains high-consistency with the real environment and exhibits good robustness throughout the entire process.

To clearly show the effects of semantic mapping, we exploited different colors to render the entities according to the results of YOLOv8-seg instance segmentation, and construct the dense semantic map and semantic octree map of the real scene, respectively. From Figure 10c,d, we find that the semantic maps constructed by our method can accurately reveal the corresponding entities, and the whole map was not misaligned and had good consistency.

Meanwhile, we utilized the SLAM trajectory evaluation tool “EVO” to evaluate the global localization accuracy of our algorithm. Figure 11 shows the trajectories generated by our algorithm and ORB-SLAM2, and it can be found that the trajectory of ORB-SLAM2 exhibited serious errors compared to ours. This is mainly due to the fact that when the robot was located at the loop-closure point for the first time, there were dynamic objects in the field-of-view and part of the static background was obscured, while, when it passed through the loop-closure point for the second time, there were no dynamic objects, which causes some impact for ORB-SLAM2 using the BoW model, leading to unsuccessful detection of the closed-loop at this point, and failing to correct the positional drift in a timely manner; whereas, we combined the semantic information and the point–lines feature that can correctly detect the closed-loop.

Furthermore, we quantitatively revealed the effectiveness of closed-loop detection using Precision–Recall (P–R) curves, and balanced “P” and “R” by adjusting the normalized similarity coefficient α∈[0.1,0.8].

Figure 12 compares the P–R curves of closed-loop detection using different features in the indoor dynamic environment. It can be observed that the Precision (0.88–0.92) and Recall (0.55–0.65) of using a single feature (ORB or LBD) were limited, while the “ORB+LBD” method significantly improved Recall, but the accuracy was still constrained by dynamic impact. In contrast, “ORB+LBD+Semantic” improved the precision while maintaining a high Recall, and forming an obvious right-up shifted P–R curve, which indicates that the method of combining instance matching to calculate the similarity can provide richer semantic information, and effectively suppresses the influence of dynamic objects on loop-closure detection.

Table 8 compares the average tracking time and detection segmentation time of each frame between our algorithm and some mainstream algorithms. We find that the average time required for detecting and segmenting each frame in our method was reduced by 87% compared to Dyna-SLAM [10], 87% compared to YOLO-SLAM [36], and 70% compared to DO-SLAM [37]. Moreover, the average tracking time of our method was reduced by 76.6% compared to DO-SLAM. Hence, we could reasonably to conclude that our method has good real-time performance while ensuring accuracy.

To further verify the performance of our algorithm in large-scale scenes, we carried out a experiment in a 27 m × 20 m outdoor corridor environment. From the global mapping results in Figure 13, we find that our method can extract rich line features in outdoor corridors, which can better assist point features in localization and mapping. In Figure 13a, the addition of line features can better reveal the structural information of the scene, while the dense map constructed in Figure 13b had high consistency with the real scene, without distortion or deformation. We could reasonably to conclude that our algorithm still exhibits good accuracy and robustness in large-scale scenes.

## 5. Conclusions

This paper proposes a vSLAM system that integrates point-line features with semantic information provided by YOLOv8-seg to address the issues of poor robustness and low positioning accuracy of traditional point-feature-based vSLAM in dynamic and low texture environments. We designed an efficient 3D line-segment extraction and modeling method, which combines 2D line-segment sampling and depth back-projection, and achieves robust 3D line-segment fitting based on RANSAC. Meanwhile, Delaunay triangulation was introduced to analyze the geometric structure between point features, and combined with the semantic segmentation information, dynamic feature points were effectively eliminated to improve the localization accuracy and the consistency of mapping in dynamic environments. In addition, we designed a loop-closure detection mechanism that integrates instance-level semantic information with geometric consistency, further improving the global consistency and robustness of the system.

Simulation studies and experimental results show that our method achieves excellent performance on multiple public datasets, demonstrating good robustness and accuracy in weak-texture and real dynamic environments. However, there is still room for further optimization in the registration of 3D line-segments and map construction in the system. The line-segment matching method may still have errors in some cases, making it difficult to effectively associate the same line-segments in different frames, resulting in duplicate mapping in the map. In addition, Delaunay triangulation based on geometric relationships is also susceptible to point-matching errors. In future work, we will strive to improve the accuracy of line-segment matching and further enhance the robustness of geometric relationship modeling to achieve more accurate and efficient line feature vSLAM systems.

## Figures and Tables

**Figure 1 sensors-25-03597-f001:**
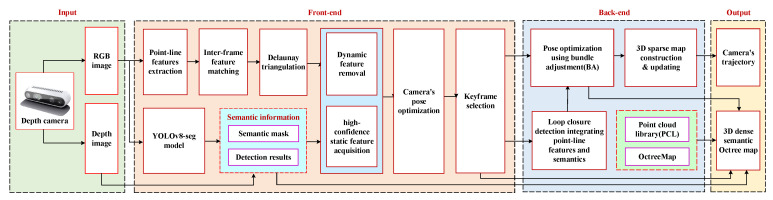
Overview of the proposed semantic visual SLAM integrating point–line features with YOLOv8-seg.

**Figure 2 sensors-25-03597-f002:**
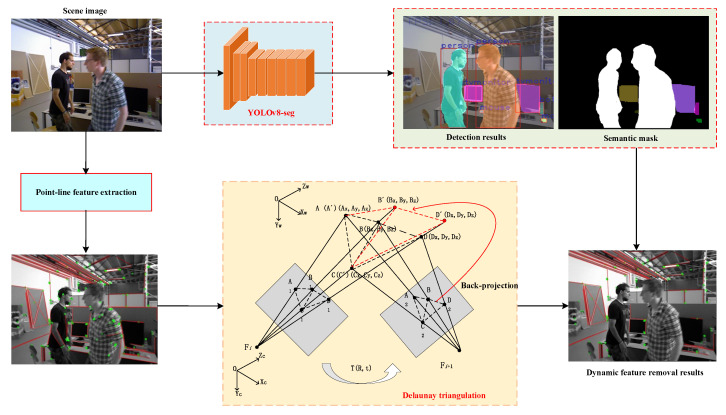
Scenarios for the dynamic feature rejection.

**Figure 3 sensors-25-03597-f003:**
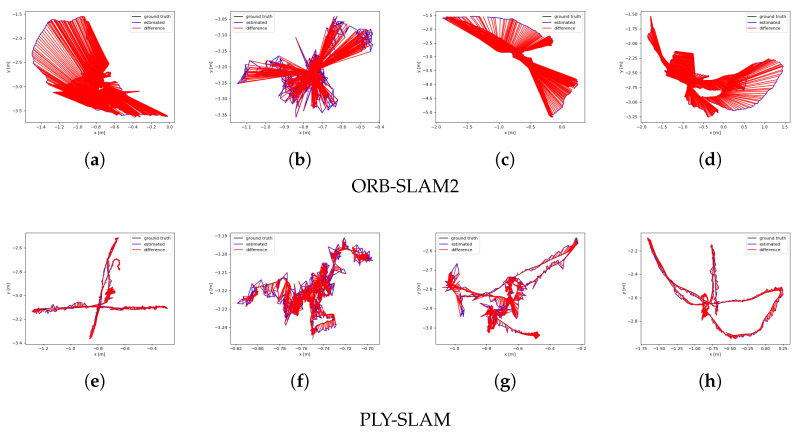
Comparison of the ATE for different algorithms under TUM dataset sequences. (**a**) fr3_walk_xyz, (**b**) fr3_walk_Static, (**c**) fr3_walk_rpy, (**d**) fr3_walk_half, (**e**) fr3_walk_xyz, (**f**) fr3_walk_Static, (**g**) fr3_walk_rpy, (**h**) fr3_walk_half.

**Figure 4 sensors-25-03597-f004:**
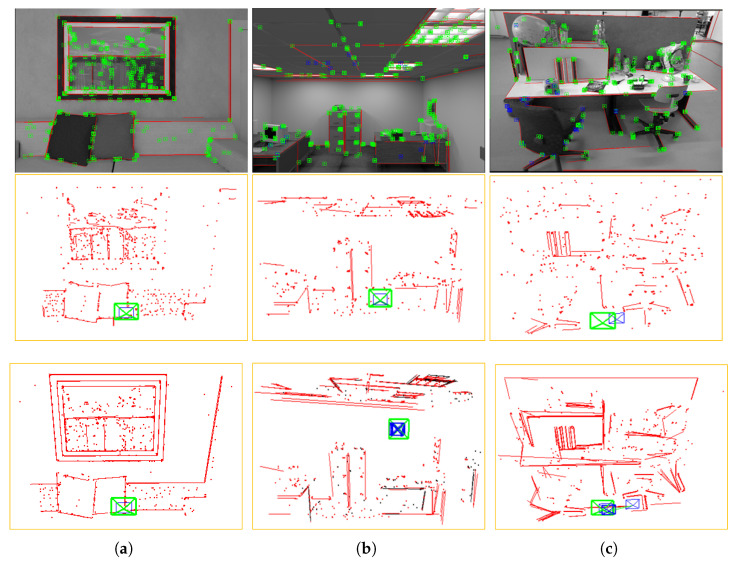
3D line-segment reconstruction results on different dataset sequences. The first row shows the ORB feature point and 2D line-segment extraction results, the middle row shows the results of triangulation of line-segment endpoints, and the bottom row shows the 3D line-segment reconstruction results.The red color indicates the extracted line segment, and the green dot indicates the extracted ORB feature point. (**a**) live_room_traj1_frei, (**b**) traj0_frei_png, (**c**) freiburg3_long_office_househol.

**Figure 5 sensors-25-03597-f005:**
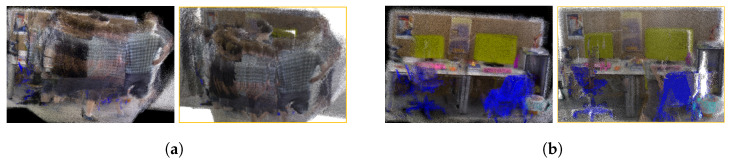
Local mapping effects of different algorithms on the dynamic fr3_walk_xyz sequence. (**a**) ORB-SLAM2, (**b**) PLY-SLAM. In (**a**,**b**), the left indicates the dense map, while the right indicates the corresponding semantic octree map.

**Figure 6 sensors-25-03597-f006:**
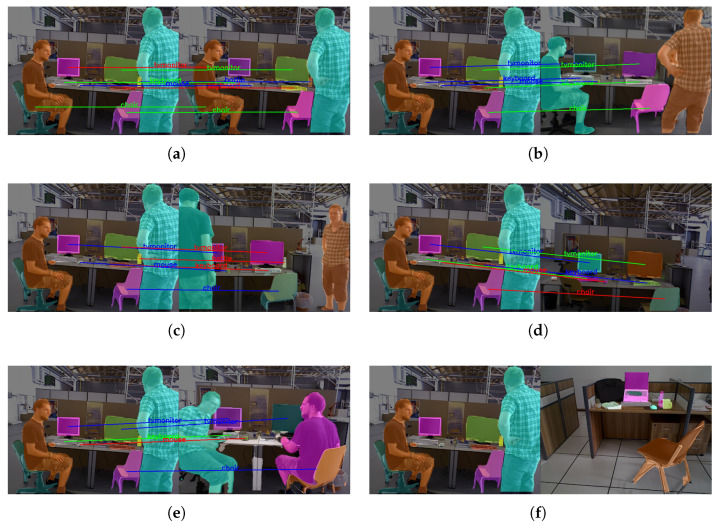
Instance-based semantic similarity-matching effects.

**Figure 7 sensors-25-03597-f007:**
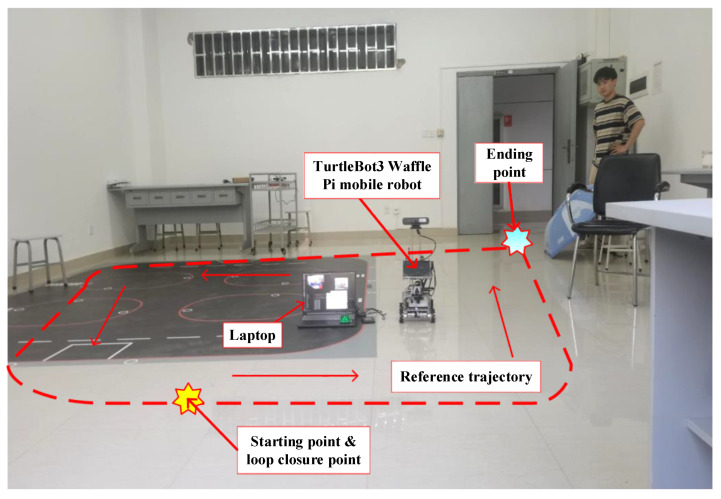
Experimental scene.

**Figure 8 sensors-25-03597-f008:**
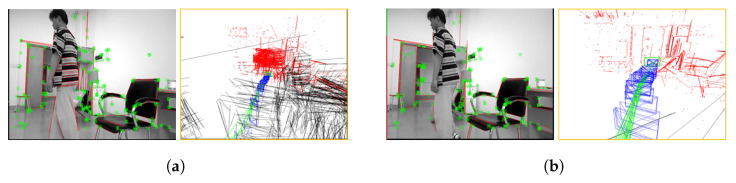
Feature extraction and sparse mapping effects in real dynamic scene. (**a**) Without removing dynamic features, (**b**) After removing dynamic features. In (**a**,**b**), the left indicates the point–line feature extraction results, while the right indicates the pose graph at the corresponding time.The red represents the 3D line segments in the current frame, the black line segments represent the 3D line segments in the local map but not in the current frame, the blue line represents the keyframe, and the green line represents the trajectory of the keyframe.

**Figure 9 sensors-25-03597-f009:**
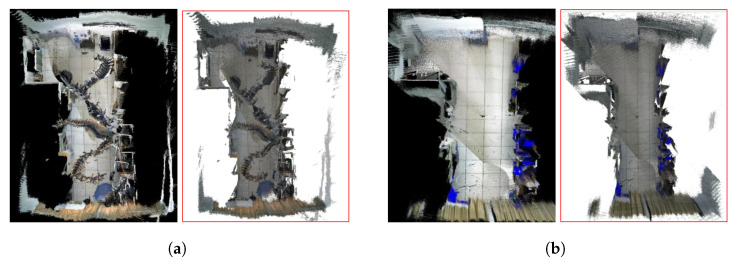
Comparison of global mapping effects of different algorithms in real dynamic environments. (**a**) ORB-SLAM2, (**b**) PLY-SLAM. In (**a**,**b**), the left indicates the semantic dense map, while the right indicates the semantic octree map.

**Figure 10 sensors-25-03597-f010:**
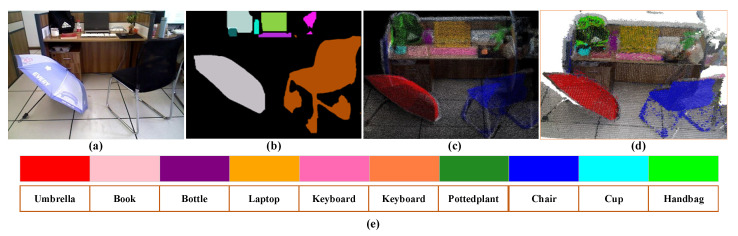
Local semantic mapping results in a real scene. (**a**) denotes the scene image, (**b**) denotes the semantic mask generated by YOLOv8, (**c**) denotes the dense semantic map, (**d**) denotes the semantic octree map, and (**e**) denotes the colors corresponding to different instances.

**Figure 11 sensors-25-03597-f011:**
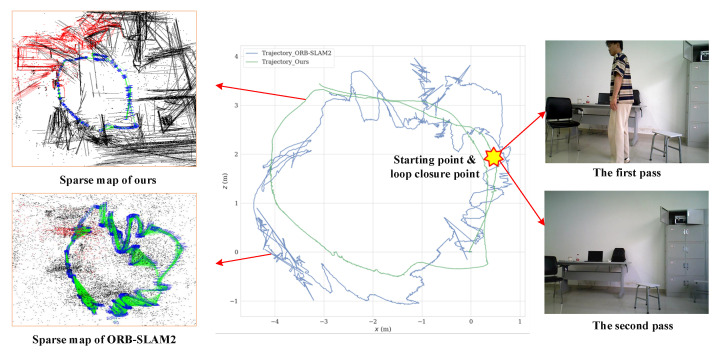
Comparison of trajectories of different algorithms in real dynamic scene.

**Figure 12 sensors-25-03597-f012:**
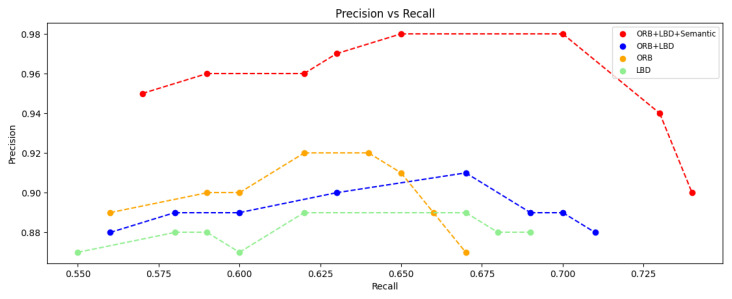
Comparison of P–R curves for loop closure detection using different feature.

**Figure 13 sensors-25-03597-f013:**
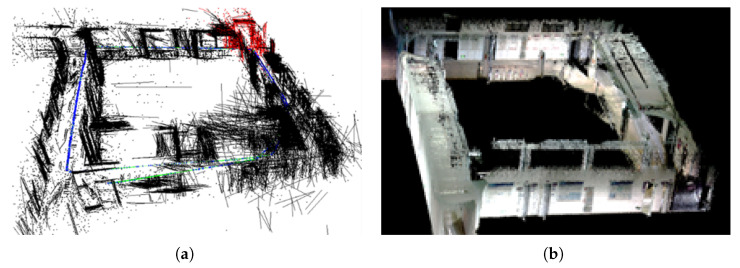
Global mapping effect in outdoor corridor environment. (**a**) is a sparse map and robot’s trajectory represented using keyframes, while (**b**) is the corresponding dense map.

**Table 1 sensors-25-03597-t001:** Comparison of RMSE for the ATE among different algorithms on TUM static dataset sequences (/m).

Sequences	PL-SLAM	RGB-DSLAM	PLP-SLAM	ORB-SLAM2	PLY-SLAM	↑PL-SLAM	↑ORB-SLAM2
fr1_xyz	0.0121	0.0116	0.0103	0.0127	0.0092	24%	27%
fr1_floor	0.0759	0.0622	0.0121	0.0390	0.0168	77%	56%
fr2_xyz	0.0043	0.0040	0.0145	0.0039	0.0035	19%	10%
fr2_large_loop	-	0.1020	-	0.1069	0.0850	-	20%
fr3_str_tex_far	0.0089	0.0092	0.0089	0.0158	0.0113	×	28%
fr3_long_office	0.0197	0.0186	0.0101	0.0126	0.0092	53%	27%
fr3_nstr_tex_n_wl	0.0206	-	0.0159	0.0212	0.0125	39%	41%
fr3_nstr_tex_far	-	0.0274	0.0352	0.0304	0.0283	-	7%

**Table 2 sensors-25-03597-t002:** Comparison of RMSE for the ATE among different algorithms on TUM dynamic dataset sequences (/m).

	Sequences	ORB-SLAM2	DS-SLAM	RDS-SLAM	DynaTm SLAM	PLY-SLAM	↑ ORB-SLAM2 (%)
Hight Dynamic	fr3_walk_xyz	0.7988	0.0274	0.0571	0.0150	0.0169	97.9
	fr3_walk_sta	0.3922	0.0081	0.0206	0.0068	0.0065	98.3
	fr3_walk_rpy	1.0799	0.4442	0.1604	0.0288	0.0364	96.6
	fr3_walk_half	0.5000	0.0303	0.0807	0.0291	0.0294	94.1
Low Dynamic	fr3_sitting_sta	0.0080	0.0065	0.0084	0.0064	0.0059	26.0

**Table 3 sensors-25-03597-t003:** Comparison of RMSE for the relative translation error (RTE) among different algorithms on TUM dataset sequences.

	Sequence	Relative Translation Error (RTE/m)					
		ORB-SLAM2	DS-SLAM	RDS-SLAM	DynaTm	PLY-SLAM	↑ORB-SLAM2 (%)
High Dynamic	fr3_walk_xyz	0.3816	0.0333	0.0426	0.0191	0.0224	94.1
	fr3_walk_sta	0.2320	0.0102	0.0221	0.0088	0.0079	96.6
	fr3_walk_rpy	0.3866	0.1503	0.1320	0.0356	0.0414	89.3
	fr3_walk_half	0.3264	0.0297	0.0482	0.0281	0.0295	91.1
Low Dynamic	fr3_sitting_sta	0.0086	0.0078	0.0123	0.0083	0.0076	11.6

**Table 4 sensors-25-03597-t004:** Comparison of RMSE for the relative rotation error (RRE) among different algorithms on TUM dataset sequences.

	Sequence	Relative Rotation Error (RRE/°)					
		ORB-SLAM2	DS-SLAM	RDS-SLAM	DynaTm	PLY-SLAM	↑ORB-SLAM2 (%)
High Dynamic	fr3_walk_xyz	7.3659	0.8266	0.9222	0.6006	0.6352	91.4
	fr3_walk_sta	4.0904	0.2690	0.4944	0.2510	0.2370	94.2
	fr3_walk_rpy	7.4997	3.0042	13.169	0.8228	0.8381	88.8
	fr3_walk_half	6.5744	0.8142	1.8828	0.7443	0.8045	87.8
Low Dynamic	fr3_sitting_sta	0.2798	0.2735	0.3338	0.2718	0.2642	5.6

**Table 5 sensors-25-03597-t005:** Average reconstruction rate of line-segments on different dataset sequences.

Dataset Sequences	Traditional Methods	PLY-SLAM
live_room_traj1_frei	51.6%	86.4%
traj0_frei_png	63.4%	84.9%
freiburg3_long_office_househol	48.7%	88.4%

**Table 6 sensors-25-03597-t006:** Calculation of similarity score.

Similarity	Figure 6a	Figure 6b	Figure 6c	Figure 6d	Figure 6e	Figure 6f
$S_{p}$	0.1087	0.0161	0.0156	0.0171	0.0139	0.0026
$S_{1}$	0.1744	0.0572	0.0356	0.0299	0.0099	0.0070
$S_{pl}$	0.1345	0.0322	0.0235	0.0221	0.0123	0.0043
$S_{inst}$	0.92	0.88	0.88	0.90	0.62	0.00
$S_{fin}$	0.4487	0.3713	0.3661	0.3733	0.2554	0.0026

**Table 7 sensors-25-03597-t007:** Main parameters of the Intel D435i depth camera.

Parameter	Properties
dimensions (length × depth × height)	90 mm × 25 mm × 25 mm
effective measurement depth	0.2∼10 m
RGB frame rate and resolution	1920 × 1080 at 30 fps
RGB sensor FOV (horizontal × vertical)	69.4° × 42.5° (+/−3°)
depth frame rate and resolution	up to 1280 × 720 at 90 fps
depth FOV (horizontal × vertical for HD 16:9)	85.2° × 58°

**Table 8 sensors-25-03597-t008:** Comparison of time consumption of different algorithms.

Algorithm	Dyna-SLAM	DO-SLAM	YOLO-SLAM	PLY-SLAM
Hardware	NVIDIA (NVIDIA Corporation, Santa Clara, CA, USA) Tesla M40 GPU	Inter Core i5-4288U	Inter Core i5-4288U CPU	Nvidia GeForce RTX 3060
Network model	Mask R-NN	YOLOv5	YOLOv3	YOLOv8-seg
Segmentation time (ms)	195	81.44	696.03	24.21
Tracking time (ms)	>300	118.23	651.53	27.61

## Data Availability

The TUM dataset was obtained from https://cvg.cit.tum.de/data/datasets/rgbd-dataset/download, accessed on 4 June 2025. ICL-NUIM dataset was obtained from https://www.doc.ic.ac.uk/~ahanda/VaFRIC/iclnuim.html, accessed on 4 June 2025.

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
