# Peer review of "PLY-SLAM: Semantic Visual SLAM Integrating Point–Line Features with YOLOv8-seg in Dynamic Scenes"

_sensors, 2025, doi:10.3390/s25123597_

Round 1
Reviewer 1 Report
Comments and Suggestions for Authors
- Table 2-4 and Figure 3 are only show the difference between the proposed method and ORB-SLAM2, but ORB-SLAM2 is not the best one. Current expression will mislead the authors to evaluate the performance among all comparative methods. It is not objective to present the performance.
- What is the computing efficiency of the proposed method? Adding segmentation will consume a large computing resources, especially at high frequency. Moreover, how the dynamic objects are defined? Only people are included?
- Figure 9. The comparison again ORB-SLAM2 is insufficient. Since this paper claims the advantage is in the plane-line features, it is more important to compare to PL-SLAM to show the performance.
- Figure 11. The trajectories and the two images on the left seem to have misalignment. Can the authors check this?
- The experiments are taken in a very small area. Moreover, room has very rich plane and line features which are beneficial for the proposed method. Therefore, more qualitative or quantitative comparisons in more complex and larger scenarios should be provided to show the outperformance.
Reviewer 2 Report
Comments and Suggestions for Authors
The paper introduces PLY-SLAM, a visual SLAM system aimed at enhancing performance in dynamic and low-texture environments. It achieves this by integrating point and line features with semantic segmentation from YOLOv8-seg. Key contributions include a novel 3D line-segment extraction technique, a hybrid geometric and semantic approach for rejecting dynamic features, and an improved loop closure mechanism that fuses point-line features with instance-level semantic matching. Experimental results on standard datasets and real-world tests show that PLY-SLAM performs favorably compared to existing methods.
Strengths
- The research directly tackles the critical issue of robust SLAM operation in challenging real-world conditions, specifically dynamic scenes and environments with limited texture.
- The paper presents evaluations on recognized datasets (TUM, ICL-NUIM) and includes real-world experiments. It uses standard metrics like Absolute Trajectory Error (ATE) and Relative Pose Error (RPE) and provides comparisons with several relevant SLAM systems.
Areas for Improvement
- Introduction and Related Work:
- While contributions are listed, the paper could more explicitly highlight how specific components of PLY-SLAM (e.g., the line extraction, dynamic feature rejection, loop closure) offer unique advantages or novel insights compared to the latest or very similar existing works. For instance, a more in-depth comparison explaining the superiority of the proposed Delaunay triangulation plus YOLOv8-seg over methods like DPL-SLAM's approach (which uses YOLOv5 and optical flow) would be beneficial, especially concerning the mentioned "over-segmentation issues".
- Methodology:
- Dynamic Feature Detection and Rejection (Section 3.2):
- Clarification is needed on how the initial dynamic weight for a feature point is determined before semantic information is applied. Is it solely based on Delaunay triangulation changes?
- The paper explains stricter conditions for potentially static objects marked as dynamic. Guidance on how these thresholds are determined (e.g., empirically, adaptively) would be valuable.
- Loop Closure Detection (Section 3.3):
- The use of information entropy to weight point and line BoW similarities is an interesting idea. A brief explanation of how this entropy reflects the "degree of influence" or discriminative power of point/line features would be beneficial.
- The mechanism for checking spatial layout by comparing neighboring instances is sound. Discussing the sensitivity or empirical tuning of the fixed reward (+0.1), punishment (*0.8), and threshold would be useful.
- Equation 24: The weights (0.6 for point-line BoW, 0.4 for instance similarity) are critical. Information on how these were determined would be valuable.
- Simulations and Experiments:
- Dataset Specifics: For the TUM RGB-D dataset, it would be helpful to specify which particular dynamic sequences (e.g., sitting_xyz, walking_xyz) were used for evaluating dynamic performance, in addition to the static sequences listed.
- Labels Used: ORB2-SLAM, ORBSLAM2, and ORB-SLAM2 are likely used to refer to the same method in this section. They are confusing for readers. Please clean them up. Instead of using "Ours", it is better to give it a meaningful name.
Round 2
Reviewer 1 Report
Comments and Suggestions for Authors
No futher questions